# Relative Entropy, Gaussian Concentration and Uniqueness of Equilibrium States

**DOI:** 10.3390/e24111513

**Published:** 2022-10-24

**Authors:** Jean-René Chazottes, Frank Redig

**Affiliations:** 1Centre de Physique Théorique, Centre National de la Recherche Scientifique, Institut Polytechnique de Paris, 91128 Palaiseau, France; 2Institute of Applied Mathematics, Delft University of Technology, 2628 CD Delft, The Netherlands

**Keywords:** lattice spin systems, translation-invariant Gibbs measure, concentration inequality, relative entropy density

## Abstract

For a general class of lattice spin systems, we prove that an abstract Gaussian concentration bound implies positivity of the lower relative entropy density. As a consequence, we obtain uniqueness of translation-invariant Gibbs measures from the Gaussian concentration bound in this general setting. This extends earlier results with a different and very short proof.

## 1. Introduction

### 1.1. Uniqueness Criteria for Gibbs Measures

In mathematical statistical mechanics, it is important to have good and useful criteria for the absence of phase transition, or equivalently, uniqueness of the Gibbs measure associated with a given potential. Such criteria, also known under the name high-temperature criteria, show that when the interaction is small enough (high temperature), there is no phase transition, and the unique phase has strong mixing properties, i.e., it is close to a product measure (infinite temperature).

The most famous among such criteria is the Dobrushin uniqueness criterion; see, e.g., ([1], Chapter 8). Under the Dobrushin uniqueness criterion, besides uniqueness, one derives strong mixing properties of the unique Gibbs measure, i.e., quantitative bounds on the decay of covariance of local observables, and quantitative bounds on the difference between finite and infinite-volume expectations, i.e., on the influence of the boundary condition on the expectation of a local function. The basic idea behind the Dobrushin uniqueness criterion is that, when it holds, the conditional expectation operator acts as a contraction on the space of probability measures equipped with the Wasserstein distance. Because the Gibbs measure is a fixed point of this contraction, and fixed points of contractions are unique, one obtains uniqueness of Gibbs measures. Later on, the Dobrushin criterion was generalized to the Dobrushin–Shlosman criterion, and a connection has been made between this criterion and an important functional inequality, the log-Sobolev inequality. More precisely, for finite-range Glauber dynamics of Ising spins, in [2] the equivalence between the Dobrushin-Shlosman criterion and the log-Sobolev inequality was proved. This implies e.g., that under the Dobrushin–Shlosman criterion, the reversible Glauber dynamics converges exponentially fast (in $L^{\infty }$) to its unique stationary measure.

Related to the Dobrushin criterion, there is a general criterion in the context of interacting particle systems under which one obtains uniqueness of the stationary measure and uniform ergodicity, i.e., from any initial measures, in time, the dynamics converges exponentially fast to the unique stationary measures. This criterion, the so-called “M<ε criterion” ([3], Chapter 1), is based on a similar contraction argument, i.e., when it holds, the semigroup of the interacting particle system acts as a contraction on a suitable space of functions, equipped with a norm (the so-called triple (semi)-norm) which controls the oscillations of a function, and the semigroup acts as a contraction in this norm. Similar to the setting of the unique measures in the setting of the Dobrushin uniqueness, also under the M<ε criterion, one obtains strong mixing properties of the unique stationary measure.

In the context of probabilistic cellular automata, as well as in the context of Glauber dynamics, the M<ε criterion is shown to be equivalent to the Dobrushin uniqueness criterion for the space–time Gibbs measure; see [4,5].

### 1.2. Concentration Inequalities

Concentration inequalities are inequalities in which one estimates the deviation between a function $f=f(\sigma_{1},\ldots ,\sigma_{n})$ of *n* random variables $\sigma_{1},\ldots ,\sigma_{n}$ and its expectation E(f). The idea is that whenever the function depends only weakly on individual variables $\sigma_{i}$, and the distribution of $\left(\sigma_{1},\ldots ,\sigma_{n}\right)$ is a product, or close to a product, then the probability that *f* deviates from E(f) becomes very small. To measure the dependence of *f* of individual coordinates, one considers, e.g., the oscillation
$$\delta_{i}f=\underset{\eta_{j}\;=\;\sigma_{j},\forall j\neq i}{\sup}\left(\;f(\sigma )-f(\eta )\right).$$

An important example of a concentration inequality is the so-called Gaussian concentration bound
(1)$$P(|\;f-E\left(f\right)|>\varepsilon )\leq 2\exp\left(-\frac{C\;\varepsilon^{2}}{\sum_{i=1}^{n}{\left(\delta_{i}f\right)}^{2}}\right),\;\varepsilon >0,$$
where C>0 is a constant which does not depend on *f*, and in particular, it does not depend on *n*. For instance, if $f\left(\sigma \right)=\frac{1}{n}\sum_{i=1}^{n}\sigma_{i}$, and $\sigma_{i}\in \left\{-1,1\right\}$, then $\delta_{i}f=2/n$, and we find the upper bound $2e^{-Cn\varepsilon^{2}/4}$ for all n≥1. The power of concentration inequalities of the type (Equation 1) is that they hold for general *f*, i.e., far beyond empirical averages.

Concentration inequalities in the context of Gibbs measures for lattice spin systems have been studied in several works. In particular, in [6] the author proves an inequality of the type (Equation 1) under the Dobrushin uniqueness criterion. See [7] for a recent overview of concentration inequalities in the context of Gibbs measures.

### 1.3. Concentration and Uniqueness

The central question of this paper is the following. Assume that a Gibbs measure associated with a given potential satisfies a Gaussian concentration bound, i.e., an inequality of the type (Equation 1). Can we then conclude that it is the unique Gibbs measure, i.e., that there is no phase transition?

In this paper, we restrict ourselves to translation-invariant Gibbs measures (i.e., so-called equilibrium states), but in a very general setting. Following [8], we started in [9] the study of the relation between the Gaussian concentration bound and the uniqueness of equilibrium states in the context of spin systems on the lattice $Z^{d}$, where the spin at each lattice site takes a finite number of values. Examples there include the Ising model at high temperature. Notice that for this model, at low temperature in d≥2, there is a phase transition, and the large deviation probabilities of the magnetization are surface-like, rather than volume-like. This manifestation of a phase transition excludes the Gaussian concentration bound, under which all ergodic averages have volume-like large-deviation probabilities.

Here, we show uniqueness of equilibrium states under an inequality of the type (Equation 1), and next, we generalize both the context of the concentration inequality, as well as the context of Gibbs measures, showing uniqueness in the context of so-called zero-information sets. An important result in the context of equilibrium states is the variational principle, which implies that the relative entropy density between two equilibrium states is zero. Therefore, if one can show a strictly positive lower bound for relative entropy density, one obtains uniqueness of the equilibrium states. The set of equilibrium states associated with a given translation-invariant potential is a special case of a set in which the relative entropy density between two elements of the set is always zero. We call such a set a zero-information set, and generalize our results of uniqueness to this context, which includes, e.g., transformations of Gibbs measures, and stationary measures of certain interacting particle systems.

### 1.4. Content and Organization of the Paper

As sketched above, we obtain a lower bound for the lower relative entropy density in terms of a natural distance between translation-invariant probability measures, reminiscent and in the spirit of the results of Bobkov and Götze [10], who proved (in a different setting) a lower bound for the relative entropy in terms of the square of the Wasserstein distance. Because we work in the thermodynamic limit on a product space and are interested in translation-invariant probability measures, there is no translation-invariant distance on the configuration space for which we can apply the Bobkov-Götze theorem. We can avoid this problem by introducing a suitable distance on the translation-invariant probability measures (rather than on configurations).

We start by proving the lower bound on the lower relative entropy density in the context of general lattice spin systems with state space $\Omega =S^{Z^{d}}$, where the single spins take values in a metric space *S* of bounded diameter. The bounded diameter property allows us to associate a quasi-local function *f* with a natural sequence of oscillations $\delta_{i}f,i\in Z^{d}$, where $\delta_{i}f$ represents the maximal influence on the function *f* of a change in the spin at site *i*. In the final section of this paper, we provide a generalization of this by allowing more abstract single-spin spaces, and more general associated sequences of oscillations.

The rest of our paper is organized as follows. In Section 2, we introduce the basic setting of lattice spin systems and important function spaces. In Section 3, we introduce the Gaussian concentration bound, the relative entropy (density), and formulate and prove our main result in the context of a single-spin space with finite diameter. In Section 4, we discuss applications of our result to zero information distance sets, including, e.g., the set of equilibrium states with regard to absolutely summable translation-invariant potentials. In Section 5, we consider a generalization by introducing an abstract sequence of oscillations, the associated Gaussian concentration bound and state, and prove the analogue of our main result in this generalized context.

## 2. Setting

### 2.1. Configuration Space and the Translation Operator

We start from a standard Borel space (S,b) with metric $d_{S}$, and we let $\mathrm{diam}\left(S\right)=\sup_{s,s^{\prime }\in S}d_{S}\left(s,s^{\prime }\right)$. (A measurable space (S,b) is said to be standard Borel if there exists a metric on *S* which makes it a complete separable metric space, and b then denotes the associated Borel σ-algebra.) In the sequel, for notational convenience, we omit the symbol b and call *S* a standard Borel space, where we always assume that the associated σ-algebra is the Borel σ-algebra b.

We assume that diam(S)<∞. Later on, in Section 5, we will show how to weaken this assumption.

This space *S* represents the “single-spin space”, i.e., we will consider lattice spin configurations in which individual “spins” take values in *S*. We denote by $\left(\Omega_{\Lambda },b_{\Lambda }\right)$ the product space $\left(S^{\Lambda },\prod_{i\in \Lambda }b_{i}\right)$, and (Ω,B) stands for the lattice spin configuration space $\left(\Omega_{Z^{d}},b_{Z^{d}}\right)$. We equip this space with the product topology. Elements of Ω are called configurations. For η∈Ω, we denote by $\eta_{i}\in S$ its evaluation at site $i\in Z^{d}$. By $\sigma_{\Lambda }$ we mean an element of $\Omega_{\Lambda }$, and by $\eta_{\Lambda }\xi_{\Lambda^{c}}$, a configuration coinciding with η on Λ and with ξ on $\Lambda^{c}$. We denote by *S* the set of finite subsets of $Z^{d}$.

We denote by $\tau_{i}:Z^{d}\to Z^{d}$, $i\in Z^{d}$, the map which shifts, or translates, by *i*; that is, $\tau_{i}\left(j\right)=j+i$, $j\in Z^{d}$, and for $\Lambda \subset Z^{d}$, we write $\tau_{i}\left(\Lambda \right)=\Lambda +i=\left\{j\in Z^{d}:j=k+i,k\in \Lambda \right\}$, $\Lambda \subset Z^{d}$. We define the translation operator acting on configurations as follows (and use the same symbol). For each $i\in Z^{d}$, ${\left(\tau_{i}\sigma \right)}_{j}=\sigma_{j-i}$, for all $j\in Z^{d}$. This corresponds to translating σ forward by *i*. We denote by the same symbol the translation operator acting on a function f:Ω→R. For each $i\in Z^{d}$, $\tau_{i}f$ is the function defined as $\tau_{i}f\left(\sigma \right)=f\left(\tau_{i}\sigma \right)$. A (Borel) probability measure on Ω is translation invariant if, for all B∈B and for all $i\in Z^{d}$, we have $\mu \left(\tau_{i}B\right)=\mu \left(B\right)$.

We denote by $P_{\tau }\left(\Omega \right)$ the set of translation-invariant probability measures on Ω. We denote by $C\left(\Omega \right),C_{b}\left(\Omega \right)$ the space of continuous, respectively bounded continuous, real-valued functions on Ω.

### 2.2. Local Oscillations and Function Spaces

To a continuous function f:Ω→R we associate a “sequence” of “local oscillations”, $\delta f:={\left(\delta_{i}f\right)}_{i\in Z^{d}}$, defined via
(2)$$\delta_{i}f=\underset{\begin{array}{c} \sigma ,\;\eta \;\in \;\Omega :\\ \sigma_{j}\;=\;\eta_{j},\forall j\neq i \end{array}}{\sup}\left(\;f(\sigma )-f(\eta )\right).$$

Later on, in Section 5, in which we consider the case where *S* is allowed to have infinite diameter, we will consider a more abstract definition of δf. In the case where *S* has finite diameter, (Equation 2) is the most natural choice.

For an integer p≥1, we define the usual $\ell^{p}$-norm of δf, ${∥\delta f∥}_{p}^{p}=\sum_{\;i\;\in Z^{d}}{\left(\delta_{i}f\right)}^{p}$, and ${∥\delta f∥}_{\infty }=\sup_{i\;\in Z^{d}}\delta_{i}f$.

We call a continuous function local if $\delta_{i}f\neq 0$ for finitely many $i\in D_{f}\subset Z^{d}$. The set $D_{f}$ is then called the dependence set of *f*. We denote by L(Ω) the set of local continuous functions on Ω.

We call a continuous function quasi-local if it is the uniform limit of a sequence of local continuous functions. If *S* is compact, then, according to the Stone–Weierstrass theorem, local continuous functions are uniformly dense in C(Ω).

We denote by QL(Ω) the space of all continuous quasi-local functions on Ω.

For 1≤p≤∞, we introduce the spaces
$$\Delta^{p}\left(\Omega \right)={\{\;f\in C\left(\Omega \right):∥\delta f∥}_{p}<\infty \}.$$

**Lemma** **1.**
*If $f\in \mathrm{QL}\left(\Omega \right)\cap \Delta^{1}\left(\Omega \right)$, then f is bounded. If $f\in L\left(\Omega \right)\cap \Delta^{p}\left(\Omega \right)$ for some p≥1, then f is bounded.*


**Proof.** Choose η,σ∈Ω. We have for every Λ∈S
(3)$$|\;f\left(\eta \right)-f\left(\sigma \right)|\leq |\;f\left(\eta \right)-f\left(\eta_{\Lambda }\sigma_{\Lambda^{c}}\right)|+\sum_{i\;\in \Lambda }\delta_{i}f$$
which, upon taking the limit $\Lambda ↑Z^{d}$, using the assumed quasilocality of *f* gives
(4)$$\left|\;f\left(\eta \right)-f\left(\sigma \right)\right|\leq \sum_{i\;\in Z^{d}}\delta_{i}f={∥\delta f∥}_{1}<\infty .$$
If *f* is local, then we still have the inequality (Equation 3) for Λ containing the dependence set of *f*. Because, by assumption, $\sum_{i}{\left(\delta_{i}f\right)}^{p}$ is finite, it follows that $\delta_{i}f<\infty$ for all i∈Λ, and therefore ${∥\delta f∥}_{1}<\infty$. Then, we obtain (Equation 4), which implies that *f* is bounded. □

We say that $\mu_{n}\to \mu$ if, for all bounded continuous local functions, we have $\int f\;d\mu_{n}\to \int f\;d\mu$ (then by definition of quasilocality, the same holds for bounded continuous quasi-local functions). This induces the so-called weak quasi-local topology on probability measures. Notice that in our setting, where by assumption the single-spin space *S* is a complete separable metric space, this topology coincides with the ordinary weak topology; see [11] (p. 898).

In our setting, the set of bounded quasi-local continuous functions is measure separating, i.e., for two probability measures $\mu \neq \mu^{\prime }$, there exists a bounded quasi-local continuous *f*, such that
$$\int f\;d\mu \neq \int f\;d\mu^{\prime }.$$
Because, by definition, bounded continuous quasi-local functions can be uniformly approximated by bounded continuous local functions, if the set of bounded quasi-local functions is measure separating, then the set of bounded continuous local functions is also measure separating. Therefore, in our setting, for two probability measures $\mu \neq \mu^{\prime }$, there exists a bounded local *f* (which is not constant), such that
$$\int f\;d\mu \neq \int f\;d\mu^{\prime }.$$
This can be seen as follows. If $\mu \neq \mu^{\prime }$, then there exists a bounded closed cylindrical set A⊂Ω, such that $\mu \left(A\right)\neq \mu^{\prime }\left(A\right)$, because the Borel σ-algebra on Ω is generated by such sets. The indicator of this set can be approximated by bounded local continuous functions in both $L^{1}\left(\mu \right)$ and $L^{1}\left(\mu^{\prime }\right)$.

## 3. Gaussian Concentration Bound and Relative Entropy

### 3.1. Abstract Gaussian Concentration Bound

We can now give the definition of the Gaussian concentration bound in our setting.

**Definition** **1.**
*Let $\Omega =S^{Z^{d}}$, where S is a standard Borel space with a finite diameter. Let μ be a probability measure on Ω. We say that μ satisfies the Gaussian concentration bound with constant C>0, abbreviated $\mathrm{GCB}\left(C\right)$, if for all bounded local functions f we have*

(5)
$$\int e^{\;f-\int f\;d\mu }\;d\mu \leq e^{\frac{C}{2}{∥\delta f∥}_{2}^{2}}.$$



**Remark** **1.**
(a)
*Observe that the bound (Equation 5) does not change if f is replaced by f+c, where c∈R is arbitrary, since $\delta_{i}\left(f+c\right)=\delta_{i}\left(f\right)$ for any $i\in Z^{d}$. This “insensitivity” to constant offsets on the left-hand side is ensured by the fact that we center f around its expected value. We also observe that (Equation 5) is trivially true for functions which are constant.*
(b)
*We have $\delta_{i}\left(\beta f\right)=\left|\beta \right|\;\delta_{i}f$, for all $i\in Z^{d}$ and β∈R; we thus have*

$$\log\int e^{\;\beta (\;f-\int f\;d\mu )}\;d\mu \leq \frac{C\beta^{2}}{2}{∥\delta f∥}_{2}^{2},\;\forall \beta \in R.$$



*This quadratic upper bound will be crucial in the sequel.*

(c)
*The quadratic nature of the upperbound in (Equation 5) resembles the quadratic upperbound for the pressure in [12], Theorem 1.1, Equation (2.7), in terms of the Dobrushin norm. This suggests that in the Dobrushin uniqueness regime, the quadratic bound which is obtained from (Equation 5) might also be obtainable from this result. However, the Gaussian concentration inequality does not require the Dobrushin uniqueness condition; the latter is sufficient, but not necessary.*



The following proposition asserts that (Equation 5) automatically extends to a wider class of functions.

**Proposition** **1**(Self-enhancement of GCB). *Suppose that (Equation 5) holds for all bounded local f. Then, it holds for all $f\in \mathrm{QL}\left(\Omega \right)\cap \Delta^{2}\left(\Omega \right)$.*

**Proof.** By assumption, for a fixed ξ∈Ω, $f\in \mathrm{QL}\left(\Omega \right)\cap \Delta^{2}\left(\Omega \right)$ can be uniformly approximated by the local functions
$$f_{\Lambda ,\;\xi }\left(\eta \right)=f\left(\eta_{\Lambda }\xi_{\Lambda^{c}}\right).$$
By definition (Equation 2), $\delta_{i}f_{\Lambda ,\;\xi }$ is non-decreasing when Λ grows, and is bounded by $\delta_{i}f$. According to Lemma 1, it follows that $f_{\Lambda ,\;\xi }$ is bounded. Therefore $∥\delta f_{\Lambda ,\;\xi }{∥}_{2}$ is bounded by ${∥\delta f∥}_{2}$, which is finite because $f\in \Delta^{2}\left(\Omega \right)$. Therefore, using the assumed uniform convergence of $f_{\Lambda ,\;\xi }$ to *f*, and the assumed bound (Equation 5) for bounded local functions in $\Delta^{2}\left(\Omega \right)$, we obtain
$$\int e^{\;f-\int f\;d\mu }\;d\mu =\lim_{\Lambda ↑Z^{d}}\int e^{\;f_{\Lambda ,\;\xi }\;-\int f_{\Lambda ,\;\xi }\;d\mu }\;d\mu \leq \lim_{\Lambda ↑Z^{d}}e^{\frac{C}{2}{∥\delta f_{\Lambda ,\;\xi }∥}_{2}^{2}}=e^{\frac{C}{2}{∥\delta f∥}_{2}^{2}}.$$
Here, in the first equality, we used the uniform convergence of $f_{\Lambda ,\;\xi }$ to *f*. In the last equality, we used $\delta_{i}f_{\Lambda ,\;\xi }\leq \delta_{i}f$, and by assumption $\sum_{i}{\left(\delta_{i}f\right)}^{2}<\infty$, so by dominated convergence applied to the counting measure on $Z^{d}$, we have
$$\lim_{\Lambda ↑Z^{d}}\sum_{i\in Z^{d}}{\left(\delta_{i}f_{\Lambda ,\;\xi }\right)}^{2}={∥\delta f∥}_{2}^{2}.$$□

### 3.2. Relative Entropy

For a probability measure μ, we denote by $\mu_{\Lambda }$ its restriction to the sub-σ-algebra $B_{\Lambda }=\sigma \left\{\eta_{i},i\in \Lambda \right\}$, generated by the projection $p_{\Lambda }:\Omega \to \Omega_{\Lambda }$. We also denote by $B_{\Lambda }$ the set of bounded $B_{\Lambda }$-measurable functions from Ω to R.

For two probability measures $\mu ,\mu^{\prime }$ on Ω and Λ∈S, we define the relative entropy of $\mu^{\prime }$ with respect to μ by
$$S_{\Lambda }\left(\mu^{\prime }|\mu \right)=\left\{\begin{array}{cc} \int \;d\mu_{\Lambda }^{\prime }\log\frac{\;d\mu_{\Lambda }^{\prime }}{\;d\mu_{\Lambda }} & \mathrm{if}\;\mu_{\Lambda }^{\prime }\ll \mu_{\Lambda }\\ +\infty & \mathrm{otherwise}. \end{array}\right.$$
We further denote by ${\left(\Lambda_{n}\right)}_{n\in N}$ the sequence of “cubes” $\Lambda_{n}={\left[-n,n\right]}^{d}\cap Z^{d}$, n≥1.

**Definition** **2**(Lower relative entropy density). *For two probability measures $\mu ,\mu^{\prime }$ on
Ω, we define the lower relative entropy density by*
$$S_{*}\left(\mu^{\prime }|\mu \right)=\underset{n\to +\infty }{lim inf}\frac{S_{\Lambda_{n}}\left(\mu^{\prime }|\mu \right)}{|\Lambda_{n}|}.$$

We have the following variational characterization of relative entropy (for proof, see, for instance, [13] (p. 100))
(6)$$S_{\Lambda }\left(\mu^{\prime }|\mu \right)=\underset{f}{\sup}\left(\;\int f\;d\mu_{\Lambda }^{\prime }-\log\int e^{\;f}\;d\mu_{\Lambda }\right)$$
where the supremum is taken over all $B_{\Lambda }$-measurable functions, such that $\int e^{\;f}\;d\mu_{\Lambda }<\infty$.

### 3.3. Main Result

In the main theorem below, we prove that the Gaussian concentration bound implies strict positivity of the lower relative entropy density. Introducing an appropriate metric on the set of probability measures, we show that the lower relative entropy density is lower bounded by a constant multiplied by the square of this distance. This result substantially generalizes the corresponding result from [9], where it is essential that the single-spin space is finite. Moreover, the proof is simpler and based on the variational formula for the relative entropy, combined with a quadratic estimate for the log-moment-generating function coming from the assumed Gaussian concentration bound.

**Definition** **3.**
*Define the following distance between probability measures*

(7)
$$d\left(\mu ,\mu^{\prime }\right)=\sup\left\{\int f\;d\mu -\int f\;d\mu^{\prime }:f\in L\left(\Omega \right),{∥\delta f∥}_{1}\leq 1\right\}.$$



The metric defined above generates the quasi-local topology, and therefore convergence in this metric implies weak convergence. Indeed, convergence $\mu_{n}\to \mu$ in the metric *d* clearly implies $\int f\;d\mu_{n}\to \int f\;d\mu$ for all local continuous *f*, and hence also for all quasi-local continuous *f*. The latter implies $\mu_{n}\to \mu$ in the quasi-local topology, which coincides with the weak topology.

We can then formulate our main result.

**Theorem** **1.**
*If μ is translation invariant and satisfies $\mathrm{GCB}\left(C\right)$, then for all $\mu^{\prime }$ translation invariant, and $\mu^{\prime }\neq \mu$, we have*

$$S_{*}\left(\mu^{\prime }|\mu \right)>0.$$

*More precisely, we have*

(8)
$$S_{*}\left(\mu^{\prime }|\mu \right)\geq \frac{d{\left(\mu^{\prime },\mu \right)}^{2}}{2C},$$

*where d is the distance (Equation 7).*


We start with a lemma from [7]. For the reader’s convenience, we repeat the short proof here.

**Lemma** **2.**
*For f, such that ${∥\delta f∥}_{1}<+\infty$ and Λ∈S, we have*

$${∥\delta \left(\;\sum_{i\;\in \Lambda }\tau_{i}f\right)∥}_{2}^{2}\leq \left|\Lambda \right|{∥\delta f∥}_{1}^{2}.$$



**Proof.** For $\Lambda \subset Z^{d}$, let ${𝟙}_{\Lambda }$ denote the indicator function of Λ (that is, ${𝟙}_{\Lambda }\left(i\right)=1$ if i∈Λ and ${𝟙}_{\Lambda }\left(i\right)=0$ otherwise). Then, for every $j\in Z^{d}$ we have
$$0\leq \delta_{j}\left(\;\sum_{i\;\in \Lambda }\tau_{i}f\right)\leq \sum_{i\;\in \;Z^{d}}\left(\delta_{i+j}f\right){𝟙}_{\Lambda }\left(i\right)={\left(\delta f*{𝟙}_{\Lambda }\right)}_{j}.$$
As a consequence, using Young’s inequality for convolutions, we obtain
$${∥\delta \left(\;\sum_{i\;\in \Lambda }\tau_{i}f\right)∥}_{2}^{2}\leq ∥\delta f*{𝟙}_{\Lambda }{∥}_{2}^{2}\leq ∥{𝟙}_{\Lambda }{∥}_{2}^{2}{\;∥\delta f∥}_{1}^{2}={|\Lambda |∥\delta f∥}_{1}^{2}.$$□

**Proof** **of** **Theorem** **1.**For the cube $\Lambda_{n}$ and a bounded local function *f* whose dependence set is included in the cube $\Lambda_{r}$, for some *r*, it follows from (Equation 6) that
$$\frac{S_{\Lambda_{n+r}}\left(\mu^{\prime }|\mu \right)}{|\Lambda_{n}|}\geq \frac{1}{|\Lambda_{n}|}\left(\int \sum_{i\in \Lambda_{n}}\tau_{i}f\;d\mu^{\prime }-\log\int e^{\;\sum_{i\in \Lambda_{n}}\tau_{i}f}\;d\mu \right)$$
where we used that $\sum_{i\in \Lambda_{n}}\tau_{i}f$ is measurable with respect to $B_{\Lambda_{n+r}}$. Now, if μ satisfies $\mathrm{GCB}\left(C\right)$ and both $\mu^{\prime }$ and μ are translation invariant, then we can estimate further as follows. Start by noticing that, through combination of the assumed $\mathrm{GCB}\left(C\right)$ and Lemma 2, we have
$$\log\int e^{\;\sum_{i\in \Lambda_{n}}\left(\tau_{i}f-\int f\;d\mu \right)}\;d\mu \leq \frac{C}{2}|\Lambda_{n}|{∥\delta f∥}_{1}^{2}.$$
As a consequence, using translation invariance of both μ and $\mu^{\prime }$, we obtain
(9)$$\begin{array}{cc} \; & \frac{S_{\Lambda_{n+r}}\left(\mu^{\prime }|\mu \right)}{|\Lambda_{n}|}\\ \; & \geq \frac{1}{|\Lambda_{n}|}\left(\int \sum_{i\;\in \Lambda_{n}}\tau_{i}f\;d\mu^{\prime }-\log\int e^{\;\sum_{i\in \Lambda_{n}}\tau_{i}f}\;d\mu \right)\\ \; & =\frac{1}{|\Lambda_{n}|}\left(\int \sum_{i\;\in \Lambda_{n}}\tau_{i}f\;d\mu^{\prime }-\int \sum_{i\;\in \Lambda_{n}}\tau_{i}f\;d\mu -\log\int e^{\;\sum_{i\;\in \Lambda_{n}}\left(\tau_{i}f-\int \tau_{i}f\;d\mu \right)}\;d\mu \right)\\ \; & \geq \int f\;d\mu^{\prime }-\int f\;d\mu -\frac{C}{2}{∥\delta f∥}_{1}^{2}. \end{array}$$
Consider a bounded local function *f*, such that $\int f\;d\mu^{\prime }-\int f\;d\mu \geq u>0$ (this function exists by the assumption that bounded local functions are measure separating). Put ${∥\delta f∥}_{1}^{2}=:\varrho$. (Observe that ϱ<∞ by assumption, and ϱ≠0 since *f* cannot be a constant.) Assume that the dependence set of *f* is included in the cube $\Lambda_{r}$. Replace *f* by βf in the inequality (Equation 9), and optimize over β. Then, we obtain, for all n∈N, the inequality
(10)$$\begin{array}{ccc} \frac{S_{\Lambda_{n+r}}\left(\mu^{\prime }|\mu \right)}{|\Lambda_{n}|} & \geq & \underset{\beta \geq 0}{\sup}\left\{\beta \left(\int f\;d\mu^{\prime }-\int f\;d\mu \right)-\frac{C}{2}\beta^{2}{∥\delta f∥}_{1}^{2}\right\}\\ & \geq & \underset{\beta \geq 0}{\sup}\left\{\beta u-\frac{C}{2}\beta^{2}\varrho \right\}=\frac{u^{2}}{2\;C\varrho }>0. \end{array}$$
Since *r* is fixed, we can take the limit inferior in *n*, and using $|\Lambda_{n}\left|/\right|\Lambda_{n+r}|\to 1$ as n→∞, we obtain
$$\underset{n\to +\infty }{lim inf}\frac{S_{\Lambda_{n}}\left(\mu^{\prime }|\mu \right)}{|\Lambda_{n}|}>0.$$
From (Equation 10), we infer that for *f*, such that
$$\int f\;d\mu^{\prime }-\int f\;d\mu \geq u,\;\mathrm{and}\;{∥\delta f∥}_{1}^{2}\leq \varrho$$
we have
$$S_{*}\left(\mu^{\prime }|\mu \right)\geq \frac{1}{2C}{\left(\frac{u}{\sqrt{\varrho }}\right)}^{2}.$$
Therefore, for
$$\int f\;d\mu^{\prime }-\int f\;d\mu \geq \varepsilon \;\mathrm{and}\;{∥\delta f∥}_{1}^{2}\leq 1$$
we have
(11)$$S_{*}\left(\mu^{\prime }|\mu \right)\geq \frac{\varepsilon^{2}}{2C}.$$
By definition of the distance (Equation 7), this is equivalent with the statement that $d(\mu^{\prime },\mu )\geq \varepsilon$ implies (Equation 11). This implies (Equation 8). □

The following corollary shows that convergence in relative entropy density implies convergence in the distance *d*. This can be used for stochastic dynamics, provided one can show that the relative entropy density converges. See the application section below for some examples.

**Corollary** **1.**
*Let μ be a translation-invariant probability measure which satisfies $\mathrm{GCB}\left(C\right)$. Assume ${\left(\mu_{n}\right)}_{n\geq 1}$ is a sequence of translation-invariant probability measures, such that*

(12)
$$\lim_{n\to +\infty }S_{*}\left(\mu_{n}|\mu \right)=0.$$

*Then, $\mu_{n}\to \mu$ in the sense of the distance (Equation 7), and therefore also $\mu_{n}\to \mu$ weakly.*


**Proof.** By (Equation 8), (Equation 12) implies
$$\lim_{n\to +\infty }d{\left(\mu_{n},\mu \right)}^{2}\leq 2C\lim_{n\to +\infty }S_{*}\left(\mu_{n}|\mu \right)=0.$$
Therefore, we have convergence in the metric *d*, which, as we remarked before, implies weak convergence. □

**Remark** **2.**
*As an example of application of Corollary (1), we mention the iteration of renormalization group transformations in the high-temperature regime [14], where convergence of the renormalized potentials can be established, and as a consequence, we obtain convergence of the relative entropy density. Then, Corollary (1) implies that the renormalized measures converge in the metric d as least as fast as the potentials. In the context of stochastic dynamics, i.e., where $\mu_{n}$ is a time-evolved measure (at time n), it is usually not simple to obtain the convergence $S_{*}\left(\mu_{n}|\mu \right)\to 0$. In the high-temperature setting (high-temperature dynamics, high-temperature initial measure) this can be obtained with similar means as in [14].*


We conclude this section with two further remarks relating our result to the Bobkov-Götze criterion.

**Remark** **3.**
*Our distance $d(\mu ,\mu^{\prime })$ between probability measures resembles the so-called Dobrushin distance, denoted by $D(\mu ,\mu^{\prime })$, which consists of taking the supremum of $\int f\;d\mu -\int f\;d\mu^{\prime }$ over a wider set of functions. Namely, f is required to be measurable, and such that ${∥\delta f∥}_{1}\leq 1$. Hence $d\left(\mu ,\mu^{\prime }\right)\leq D\left(\mu ,\mu^{\prime }\right)$ for a general pair $\mu ,\mu^{\prime }$ of probability measures. In the special case of finite S, one has d=D. In [15], it is proved that D is equal to what the authors called the Steiff distance $\bar{d}$, which is defined in terms of couplings, and which generalizes the Ornstein distance. The equality between D and $\bar{d}$ is reminiscent of the Kantorovich–Rubinstein duality theorem.*


**Remark** **4.**
*Inequality (Equation 8) is reminiscent of a well-known abstract inequality relating the relative entropy and the Wasserstein distance due to Bobkov and Götze [10]. However, our context is different, because we consider the thermodynamic limit and the relative entropy density. Nevertheless, as shown in [7], we can exploit the Bobkov-Götze theorem in the special case of finite S, putting the Hamming distance on $S^{\Lambda_{n}}$, to get*

$$S_{*}\left(\nu |\mu \right)\geq \frac{{\bar{d}}^{\;2}\left(\mu ,\nu \right)}{2C},$$

*where*

$$\bar{d}\left(\mu ,\nu \right):=\lim_{n\to +\infty }\frac{W_{1}\left(\mu_{\Lambda_{n}},\nu_{\Lambda_{n}};{\bar{d}}_{n}\right)}{|\Lambda_{n}|}$$

*and where ${\bar{d}}_{n}\left(\omega ,\eta \right)=\sum_{\;i\in \Lambda_{n}}{𝟙}_{\left\{\omega_{i}\neq \eta_{i}\right\}}$, $W_{1}\left(\mu_{\Lambda_{n}},\nu_{\Lambda_{n}};{\bar{d}}_{n}\right)$ is the Wassertein distance between $\mu_{\Lambda_{n}}$ and $\nu_{\Lambda_{n}}$.*


## 4. Applications: Uniqueness of Equilibrium States and Beyond

In this subsection, we provide some settings where we can conclude uniqueness of a set of “(generalized) translation-invariant Gibbs measures” via Theorem 1. We start with the set of translation-invariant Gibbs measures associated with an absolutely summable potential. Then, we consider generalizations and modifications of such sets.

### 4.1. Uniqueness of Equilibrium States

In this subsection, we briefly introduce the necessary basics of Gibbs measures. The reader familiar with the theory of Gibbs measure can skip this subsection. The reader is referred to [1] (especially Chapter 16) or [11] (Chapter 2) for more background on the Gibbs formalism.

Let λ be a probability measure on *S*, and for $\lambda_{\Lambda }\left(\;d\sigma_{\Lambda }\right)={⊗}_{i\in \Lambda }\lambda \left(\;d\sigma_{i}\right)$ the corresponding product measure on $S^{\Lambda }$. The measure λ is called the “a priori” measure on *S*, with associated a priori measure ${⊗}_{i\in Z^{d}}\lambda \left(\;d\sigma_{i}\right)$ on Ω.

We call a uniformly absolutely summable translation-invariant potential a function
U:S×Ω→R
with the following properties:(a)Locality: for all A∈S, U(A,·) is $B{\;}_{A}$-measurable and continuous.(b)Absolute summability: $\sum_{A∋0}{∥U\left(A,\cdot \right)∥}_{\infty }<\infty$.(c)Translation invariance: $U\left(A+i,\tau_{i}\sigma \right)=U\left(A,\sigma \right)$ for all σ∈Ω, A∈S.

Let us call U the set of uniformly absolutely summable translation-invariant potentials. Then, we build the local Gibbs measures with boundary condition ξ∈Ω. For a finite subset Λ∈S, the Gibbs measure in volume Λ with boundary condition ξ outside Λ is defined via
$$\gamma_{\Lambda }\left(\;d\sigma_{\Lambda }|\xi \right)=\frac{e^{-H_{\Lambda }^{\;\xi }\left(\sigma_{\Lambda }\right)}}{Z_{\Lambda }^{\xi }}\lambda_{\Lambda }\left(\;d\sigma_{\Lambda }\right)$$
where $H_{\Lambda }^{\xi }$ is the Hamiltonian in volume Λ with boundary condition ξ:$$H_{\Lambda }^{\;\xi }\left(\sigma_{\Lambda }\right)=\sum_{A\cap \Lambda \neq \emptyset }U\left(A,\sigma_{\Lambda }\xi_{\Lambda^{c}}\right)$$
and where $Z_{\Lambda }^{\xi }$ is the normalization
$$Z_{\Lambda }^{\xi }=\int e^{-H_{\Lambda }^{\;\xi }\left(\sigma_{\Lambda }\right)}\lambda_{\Lambda }\left(\;d\sigma_{\Lambda }\right).$$
The family ${\left(\gamma_{\Lambda }\left(\;d\sigma_{\Lambda }|\cdot \right)\right)}_{\Lambda \in S}$ is called the Gibbsian specification associated with the potential *U* (with a priori measure λ).

By the uniform absolute summability of *U*, we automatically have that for all *f* local and continuous, the function $\xi \mapsto \int f\left(\sigma_{\Lambda }\xi_{\Lambda^{c}}\right)\;\gamma_{\Lambda }\left(\;d\sigma_{\Lambda }|\xi \right)$ is quasi-local and continuous. We say that the specification $\gamma_{\Lambda }\left(\cdot |\cdot \right)$ is quasi-local.

We then call a measure Gibbs μ with potential *U* (and a priori measure λ) if $\gamma_{\Lambda }\left(\;d\sigma_{\Lambda }|\xi \right)$ is consistent with the finite-volume Gibbs measures, i.e., if for all f:Ω→R bounded and measurable, and Λ∈S, we have
$$E_{\mu }\left(\;f|B_{\Lambda^{c}}\right)\left(\xi \right)=\int f\left(\sigma_{\Lambda }\xi_{\Lambda^{c}}\right)\;\gamma_{\Lambda }\left(\;d\sigma_{\Lambda }|\xi \right)$$
for μ-almost every ξ.

We denote by $G_{\tau }\left(U\right)$ the set of translation-invariant Gibbs measures associated with the potential *U*. These measures are called the “equilibrium states” associated with *U*.

The variational principle ([1] Chapter 16) implies that if $\mu ,\nu \in G_{\tau }\left(U\right)$, then $s_{*}\left(\mu |\nu \right)=s_{*}\left(\nu |\mu \right)=0$, and conversely if $\mu \in G_{\tau }\left(U\right)$ and $\nu \in P_{\tau }$, is such that $s_{*}\left(\nu |\mu \right)=0$, then $\nu \in G_{\tau }\left(U\right)$. As a consequence of Theorem 1, we then obtain the following result:

**Proposition** **2.***Let*U∈U*. If*$\mu \in G_{\tau }\left(U\right)$*satisfies*$\mathrm{GCB}\left(C\right)$*for some*C>0*, then*$G_{\tau }\left(U\right)=\left\{\mu \right\}$.

This substantially extends the implication between $\mathrm{GCB}\left(C\right)$ and uniqueness of equilibrium states from [9], where we only considered finite single-spin spaces S.

**Remark** **5.**
*Because our techniques are based on relative entropy density, we cannot exclude the existence of non-translation-invariant Gibbs measures, even in the presence of a translation-invariant Gibbs measure satisfying $\mathrm{GCB}\left(C\right)$. In other words, even if there exists a unique equilibrium state, there might still be non-translation-invariant Gibbs measures. We believe, however, that the presence of a translation-invariant Gibbs measure satisfying $\mathrm{GCB}\left(C\right)$ implies a stronger form of uniqueness, which excludes the presence of non-translation-invariant Gibbs measures.*


### 4.2. Sets of Zero-Information Distance

The example of the set of equilibrium states from the previous subsection leads naturally to the more general notion of “zero-information distance sets” defined below.

**Definition** **4.**
*We call a subset $K\subset P_{\tau }\left(\Omega \right)$ a zero-information distance set if for all $\mu ,\mu^{\prime }\in K$, $s_{*}\left(\mu |\mu^{\prime }\right)=s_{*}\left(\mu^{\prime }|\mu \right)=0$.*


From Theorem 1, we then immediately obtain the following proposition.

**Proposition** **3.**
*Let*

$K\subset P_{\tau }\left(\Omega \right)$

*be a zero-information distance set. If there exists*

μ∈K

*, which satisfies*

$\mathrm{GCB}\left(C\right)$

*for some*

C∈(0,∞)

*, then*

K

*is a singleton.*


We provide four further examples (beyond equilibrium states) of such zero-information distance sets, illustrating Proposition 3.
(a)**Asymptotically decoupled measures and $\Pi {\;}^{f}$-compatible measures.**A first generalization of the Gibbsian context is provided in the realm of “asymptotically decoupled measures” via the notion of $\Pi {\;}^{f}$-compatible measures, see [16]. This setting goes beyond quasi-local specifications, and therefore includes many relevant examples of non-Gibbsian measures.In this setting, the set of $\Pi {\;}^{f}$-compatible measures (associated with a local function *f*) is a zero-information set ((see [16] Theorem 4.1), and therefore, if this set contains an element *μ* satisfying $\mathrm{GCB}\left(C\right)$, then it coincides with the singleton {μ}.(b)**Renormalization group transformations of Gibbs measures.**Another important class of examples is the following. We say that a transformation $T:P_{\tau }\left(\Omega \right)\to P_{\tau }\left(\Omega^{\prime }\right)$ preserves zero- information distance sets if a zero-information distance set is mapped by *T* onto a zero-information distance set. Important examples of such transformations *T* are local and translation-invariant renormalization group transformations studied in [11], Section 3.1 p 960, conditions T1-T2-T3. Examples of such transformations include block-spin averaging, decimation, and stochastic transformations such as the Kadanoff transformation. Because the transformations are “local and translation-invariant probability kernels”, one immediately infers the property $s_{*}\left(\mu T|\nu T\right)\leq s_{*}\left(\mu |\nu \right)$.In this setting, Proposition 3 implies that if U∈U, $\mu \in G_{\tau }\left(U\right)$ is an associated translation-invariant Gibbs measure, and μT satisfies $\mathrm{GCB}\left(C\right)$ for some C∈(0,∞), then ν=μT for all *ν*, such that $s_{*}\left(\nu |\mu T\right)=0$. In particular, this implies that $\mu^{\prime }T=\mu T$ for all $\mu^{\prime }\in G_{\tau }\left(U\right)$. Indeed, in that case $s_{*}\left(\mu^{\prime }T|\mu T\right)\leq s_{*}\left(\mu^{\prime }|\mu \right)=0$.Notice that μT can be non-Gibbs; therefore, the implication ν=μT for all *ν* such that $s_{*}\left(\nu |\mu T\right)=0$ cannot be derived from the variational principle.(c)**Projections of Gibbs measures.**Let *μ* be a translation-invariant Gibbs measure on the state space $S^{Z^{d}}$ (associated with a translation-invariant potential) which satisfies $\mathrm{GCB}\left(C\right)$ for some C∈(0,∞). Let for $d^{\prime }<d$, $\mu_{d^{\prime }}$ denote its restriction to the sublattice $L_{d^{\prime }}:=\left\{\left(x_{1},\ldots ,x_{d^{\prime }},0,\ldots ,0\right):x_{1},\ldots ,x_{d^{\prime }}\in Z\right\}$. It is clear that $\mu_{d^{\prime }}$ satisfies $\mathrm{GCB}\left(C\right)$ with the same constant C∈(0,∞). Therefore, any translation-invariant measure on $S^{L_{d^{\prime }}}$ that differs from $\mu_{d^{\prime }}$ has strictly positive lower relative entropy density with regard to $\mu_{d^{\prime }}$.As a consequence, if $\mu_{d^{\prime }}$ is a Gibbs measure for a translation-invariant potential $U_{d^{\prime }}^{\prime }$, then this potential $U_{d^{\prime }}^{\prime }$ has no other translation-invariant Gibbs measures. This gives uniqueness for a set of Gibbs measures where the potential is only implicitly defined, and can be complicated, i.e., uniqueness is not a consequence of a simple criterion.Projections of Gibbs measures arise naturally in the context of probabilistic cellular automata, where the stationary measures are projections of the space–time Gibbs measures [4]. In this setting, the result tells us that if the space–time measure satisfies $\mathrm{GCB}\left(C\right)$ for some C>0, then the unique stationary measure, if Gibbs, has a potential with a unique equilibrium state. Projections of Gibbs measures can fail to be Gibbs, as is shown in [17] for projection of the low-temperature Ising model in d=2 on the X-axis. It is an open and interesting problem to investigate whether this projected measure satisfies the Gaussian concentration bound.(d)**Stationary measures for Ising spin Glauber dynamics.**An additional example of a zero-information distance set is the set of stationary and translation-invariant measures for (Ising spin, i.e., *S* is finite) Glauber dynamics under the condition that this set contains at least one translation-invariant Gibbs measure as a stationary measure; see [18], Section 4. See also [19,20] for earlier results in the setting of reversible Glauber dynamics, and [21] for recent results in this spirit for more general local dynamics. As a consequence of Proposition 3, we then conclude that if there exists a translation-invariant Gibbs measure *ν* as stationary measure, and there exists a translation-invariant stationary measure *μ* satisfying $\mathrm{GCB}\left(C\right)$ for some C>0, then μ=ν coincide, and *μ* is the unique translation-invariant stationary measure. Moreover, if, in this setting, one can show that when starting the dynamics from a translation-invariant initial measure *μ* and denoting $\mu_{t}$ for the measure at time t>0, we have $S_{*}\left(\mu_{t}|\nu \right)\to 0$ as t→∞, then, from Corollary 1, we obtain that $\mu_{t}\to \nu$ as t→∞ in the sense of the distance (Equation 7).

## 5. Generalization

In the setting of Section 2.1, without the additional assumption of finiteness of the diameter of *S*, the definition of the oscillation of *f* in (Equation 2) is no longer appropriate. Indeed, it becomes natural to include unbounded functions, which makes (Equation 2) infinite. Consider, e.g., S=R, and $\Omega =S^{Z^{d}}$ equipped with a product of Gaussian measures, then the function $f\left(\eta \right)=\eta_{i}$ should be a possible choice. We consider now a general standard Borel S, which is such that for the product space $\Omega =S^{Z^{d}}$, quasi-local bounded functions are measure separating.

In order to proceed, we therefore associate with a function f:Ω→R an abstract sequence of oscillations $\delta f={\left(\delta_{i}f\right)}_{i\in Z^{d}}$ satisfying the following conditions.

**Definition** **5.**
*We say that a map $\delta :C\left(\Omega \right)\to {\left[0,\infty \right]}^{Z^{d}}$ is an allowed sequence of oscillations if the following four conditions are met.*



*1.* 
*Translation invariance: ${\left(\delta \left(\tau_{i}f\right)\right)}_{j}=\delta_{i+j}f$, $i,j\in Z^{d}$.*
*2.* 
*Non-degeneracy: $\delta_{i}f$ is zero for a function f if and only if f does not depend on the i-th coordinate, i.e., $\delta_{i}f=0$ if and only if for all η,σ such that $\eta_{j}=\sigma_{j}$ for all j≠i, f(η)=f(σ).*
*3.* 
*Monotonicity: for ξ∈Ω and f a bounded quasi-local function, we consider the local approximation of f given by*

$$f_{\Lambda ,\;\xi }\left(\eta \right)=f\left(\eta_{\Lambda }\xi_{\Lambda^{c}}\right).$$


*Then, we require that for all ξ, for all Λ and for all $i\in Z^{d}$, $\delta_{i}f_{\Lambda ,\;\xi }\leq \delta_{i}f$.*
*4.* 
*Degree one homogeneity: $\delta_{i}\left(\beta f\right)=\left|\beta \right|\delta_{i}f$ for all β∈R and for all $i\in Z^{d}$.*



Notice that Condition 3 implies that for given f, ξ, $\Lambda \subset \Lambda^{\prime }$, and $i\in Z^{d}$, we have $\delta_{i}f_{\Lambda ,\;\xi }\leq \delta_{i}f_{\Lambda^{\prime },\;\xi }$. Indeed, notice that ${\left(f_{\Lambda^{\prime },\;\xi }\right)}_{\Lambda ,\;\xi }=f_{\Lambda ,\;\xi }$ for $\Lambda \subset \Lambda^{\prime }$

The most natural example different from (Equation 2) is
$$\delta_{i}f=\underset{\begin{array}{c} \sigma ,\;\eta \;\in \;\Omega :\\ \sigma_{j}=\eta_{j},\forall j\neq i \end{array}}{\sup}\frac{\;f(\sigma )-f(\eta )}{d_{S}\left(\sigma_{i},\eta_{i}\right)}.$$
More generally, one can define
$$\delta_{i}f=\underset{\begin{array}{c} \sigma ,\;\eta \;\in \;\Omega :\\ \sigma_{j}=\eta_{j},\forall j\neq i \end{array}}{\sup}\frac{\;f(\sigma )-f(\eta )}{\psi (\sigma_{i},\eta_{i})}$$
where ψ:S×S→[0,∞) satisfies $\psi (s,s^{\prime })=0$ iff $s=s^{\prime }$.

For a given sequence of oscillations *δ*, we call a function *δ*-Lipschitz if $\sup_{i\;\in \;Z^{d}}\delta_{i}f<\infty$. We then introduce
$$\Delta^{p}\left(\Omega \right)={\{\;f\in C\left(\Omega \right):∥\delta f∥}_{p}<\infty \}.$$

**Definition** **6.**
*Let $\Omega =S^{Z^{d}}$, where S is a standard Borel space. Assume an allowed sequence of oscillations δ is given. Let μ be a probability measure on Ω. We say that μ satisfies the Gaussian concentration bound with regard to δ with constant C>0 (still abbreviated $\mathrm{GCB}\left(C\right)$), if for all bounded local functions we have*

$$\int e^{\;f-\int f\;d\mu }\;d\mu \leq e^{\frac{C}{2}{∥\delta f∥}_{2}^{2}}.$$



We then have the following analogue of Theorem 1. Because the proof follows exactly the same steps as the proof of Theorem 1, we leave it to the reader.

**Theorem** **2.**
*Assume δ is an allowed vector of oscillations. Assume that the set of bounded local δ-Lipschitz functions is measure separating. If μ is translation invariant and satisfies $\mathrm{GCB}\left(C\right)$, then for all $\mu^{\prime }$ translation invariant, and $\mu^{\prime }\neq \mu$ we have*

$$S_{*}\left(\mu^{\prime }|\mu \right)>0.$$



As a final comment, we remark that the fact that we have chosen the group $Z^{d}$ is for the sake of simplicity. We can work with more general amenable groups, as in [22].

## Data Availability

Not applicable.

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
