# Peer review of "Relative Entropy, Gaussian Concentration and Uniqueness of Equilibrium States"

_entropy, 2022, doi:10.3390/e24111513_

Round 1

Reviewer 1 Report

For comments, please see the attached file.

Reviewer 2 Report

    I found this paper on the whole reasonable, and worth publishing, I suggest some minor revisions and extra explanations. 

    1) The natural context in which the results apply are variations of the Dobrushin uniqueness regime, it seems to me. For compact spins the Dobrushin norm ||| \Phi|||= \sum_{0 \in X} |X| ||\Phi_X|| is more restrictive than the abolutely summable norm ||\Ph|| = \sum_{0 \in X}  ||\Phi+X||. 

    The pressure (free energy density) is C^2 in the Dobrushin norm-proven by Gross- , which also seems to fit the quadratic error term of the Gaussian concentration inequality. Maybe add a remark or comment?

    2) although the generalisation is towards some non-compact situations, the requirement of absolute summability which is natural for compact spins, works well  for finite diameter, but that seems to exclude models for unbounded spins, and excludes many popular exaples, (Gaussians, SOS, \Phi^4 models, e.g.). This is discussed a bit in the last section,  but some extra comments might be helpful.

    3) the conditions suggest to me that there should be "strong uniqueness" (there exists a unique Gibbs measure, which is translation invariant). 

    The conclusions that only a unique translation invariant Gibbs measure exists

    is a kind of "weak uniqueness",  and would allow for some phase transition situations like an Ising antiferromagent which has two extremal periodic Gibbs measures, or models where a quasicrystalline order would be allowed, but I would not expect  those to satisfy the GCB.

    Do the authors have some comment on this?

    p1, line 7: criterium ---criterion

          line -4: equally -- similarly

          End of p1, the relationship between Dobrushin and M < \epsilon has been elucidated by Maes and Shlosman, the triple norm involved is closely related to the Dobrushin norm, for example. 

    CMP 135, page 233  and 151, page 447, mention.

    p2, line 11: concentration -- concentration inequality. 

          line 24: restrict ---restrict ourselves

    p3, line 1: apply --- apply the

    p4, line 10 (line 130), a function can be quasilocal, even local, without being continuous, if the single-spin space is non-discrete, but still compact, like the characteristic function of a subinterval of spin values. This would be possible for XY spins or continuous Ising spins, e.g. Rephrase. 

Round 2

Reviewer 1 Report

Please find our detailed comments in the attached pdf.

Author Response

The reviewer pointed to some typos, which we fixed

Reviewer 2 Report

line 87 Spaceand -- space and

single spin ---single-spin (various places), eg lines 156, 186,260

The red norm bars should not be both inside and outside the lage norm bars in the equation above proof of Theorem 7, line 198

check the sentence in line 207

Author Response

The reviewer pointed to some typos which we fixed

We also added remark 3c related to an earlier remark of the referee, referring to an earlier result of Gross

Reviewer 3 Report

I have checked the reply by the authors, which was satisfactory. Comments by the other referees also improve the quality of the manuscript. Now I recommend the publication.

Author Response

the reviewer had no supplementary remarks